# Melatonin as a Promising Anti-Inflammatory Agent in an In Vivo Animal Model of Sepsis-Induced Rat Liver Damage

**DOI:** 10.3390/ijms25010455

**Published:** 2023-12-29

**Authors:** Daniela Benedeto-Stojanov, Vanja P. Ničković, Gordana Petrović, Andrija Rancić, Ivan Grgov, Gordana R. Nikolić, Zoran P. Marčetić, Milica R. Popović, Milan Lazarević, Katarina V. Mitić, Dušan Sokolović

**Affiliations:** 1Faculty of Medicine, University of Niš, 18000 Niš, Serbia; gpetrovicnis@gmail.com; 2COVID Hospital Kruševac, University Clinical Centre of Niš, 37000 Kruševac, Serbia; nickovicvanja@gmail.com; 3Clinic of Gastroenterohepatology, University Clinical Centre of Niš, 18000 Niš, Serbia; andrija.m.rancic@gmail.com; 4General Hospital Leskovac, Department of General Surgery with Traumatology, 16000 Leskovac, Serbia; grgovivan@gmail.com; 5Faculty of Medicine, University of Priština, 38220 Kosovska Mitrovica, Serbia; gordananikolic65@yahoo.com (G.R.N.); marceticz@hotmail.com (Z.P.M.); 6Pediatrics Clinic, Clinical Centre Priština, 38205 Gracanica, Serbia; milicavasicpopovic@gmail.com; 7Clinic for Cardiovascular and Transplant Surgery, Faculty of Medicine, University Clinical Centre of Niš, 18000 Niš, Serbia; dr_m.lazarevic@hotmail.com; 8Institute of Physiology and Biochemistry “Ivan Djaja”, Faculty of Biology, University of Belgrade, 11000 Belgrade, Serbia; katarinavmitic@yahoo.com; 9Institute for Biochemistry, Faculty of Medicine, University of Niš, 18000 Niš, Serbia; dusantsokolovic@gmail.com

**Keywords:** sepsis, lipopolysaccharide, melatonin, NF-κB, pro-inflammatory cytokines

## Abstract

Melatonin (MLT), earlier described as an effective anti-inflammatory agent, could be a beneficial adjunctive drug for sepsis treatment. This study aimed to determine the effects of MLT application in lipopolysaccharide (LPS)-induced sepsis in Wistar rats by determining the levels of liver tissue pro-inflammatory cytokines (TNF-α, IL-6) and NF-κB as well as hematological parameters indicating the state of sepsis. Additionally, an immunohistological analysis of CD14 molecule expression was conducted. Our research demonstrated that treatment with MLT prevented an LPS-induced increase in pro-inflammatory cytokines TNF-α and IL-6 and NF-κB levels, and in the neutrophil to lymphocyte ratio (NLR). On the other hand, MLT prevented a decrease in the blood lymphocyte number induced by LPS administration. Also, treatment with MLT decreased the liver tissue expression of the CD14 molecule observed after sepsis induction. In summary, in rats with LPS-induced sepsis, MLT was shown to be a significant anti-inflammatory agent with the potential to change the liver’s immunological marker expression, thus ameliorating liver function.

## 1. Introduction

Sepsis models in experimental animals could be induced by lipopolysaccharide (LPS) application, an endotoxin recognized as the most potent bacterial inductor of sepsis. Sepsis represents a systemic response where LPS interacts with host cell receptors, generating numerous pro-inflammatory cytokines: tumor necrosis factor-alpha (TNF-α), interleukin (IL)-1β, IL-6, IL-12, and inflammatory mediators (nitric oxide (NO)) [1]. Pro-inflammatory cytokines cause an early, uncontrolled hyperinflammatory response (“cytokine storm”), which is associated with a release of both pro-inflammatory and anti-inflammatory cytokines [2]. Given that the secretion of pro-inflammatory cytokines triggers the pathophysiological mechanisms in sepsis, the first attempts in the therapeutic action aimed to prevent inflammation and self-damage caused by a “storm of pro-inflammatory cytokines” [3]. However, in a small number of patients, a positive effect of anti-inflammatory therapy on the course and outcome of the disease has been described [4].

Furthermore, LPS is responsible for an exaggerated host immune response that results in septic shock. LPS affects liver cells and other tissues, producing many pro-inflammatory mediators by activating several transcription factors. One of the major targets of LPS signaling is the nuclear transcription factor of the kappa light chains of activated B cells (NF-κB) signaling pathway, a key regulator of the immune and inflammatory response [5]. Specifically, LPS-dependent activation of the inducible and mitochondrial isoforms of NO synthase has been responsible for the overproduction of NO [6], leading to the formation of peroxynitrite and the generation of hydroxyl radicals, followed by lipid peroxidation [7,8] and mitochondrial functional impairment.

Liver cells participate in a network system designed to detect, capture, and activate an adequate immune response against circulating microorganisms. For this purpose, together with hepatocytes, a large population of macrophages—Kupffer cells (KCs), neutrophils, NK cells, lymphocytes, dendritic cells, B cells, hepatic stellate cells (HSCs), and liver sinusoidal endothelial cells (LSECs)—contribute to the liver immune response during sepsis [9,10]. Clearance of bacteria from the liver is important in reducing plasma LPS levels, which lowers the severity of the immune response and improves the outcome of patients with sepsis [11]. The distribution of Toll-like receptor 4 (TLR-4), as the primary mediator of the LPS effect in the liver, is most pronounced in KCs and LSECs of the liver and less pronounced in hepatocytes [12]. In vivo research has shown that LPS initially reacts with KCs and LSECs in the sinusoids and afterward with stellate cells. It has been proven that the TLR-4 molecule is the primary signaling receptor for LPS in the presence of CD14 and LBP molecules [13].

Melatonin from an exogenous source exhibits significant beneficial effects in organisms [14]. Due to its anti-oxidative properties, melatonin can prevent oxidative stress (due to the increased formation of reactive oxygen species, ROS) and inflammation, which occurs due to the resulting oxidative damage. Besides this indirect anti-inflammatory effect, MLT modulates the NF-κB signaling pathway in inflammation and thus changes the transcription of genes coding pro-inflammatory cytokines [15]. By inhibiting the NF-κB signaling pathway, MLT reduces the synthesis of IL-1β, TNF-α, and NO, which are involved in developing inflammation in endotoxemia [16]. During liver failure in an experimental model of sepsis in rats, glucose metabolism is disturbed, transaminase activity increases, and levels of IL-1β, TNF-α, and IL-6 increase.

In contrast, the administration of melatonin to rats with sepsis leads to the normalization of these parameters [17]. A recent study proved that the immunomodulatory activity of MLT originates from TLR modulation, the reduction in NF-κB expression, and the reduction in the intensity of oxidative stress in vitro [18]. Melatonin has also been shown to inhibit elevated myeloperoxidase (MPO) activity [19], suggesting that its protective effects are partially related to the inhibition of neutrophil infiltration and the removal of mediators produced by neutrophils [20]. The inhibition of apoptosis is one of the mechanisms by which MLT prevents septic shock and reduces programmed cell death in the spleen and liver [21]. Almost all of the studies that monitored MLT-dependent anti-apoptotic activities included disease models characterized by a harmful reduction in oxygen levels (hypoxia) and ischemia events. These pathological conditions with dramatic clinical picture include the activation of several different pathways, and MLT is known to modulate most of them [22].

The present research aimed to investigate the protective effects of MLT in rats with LPS-induced sepsis by determining the tissue pro-inflammatory parameters associated with septic liver damage. An immunohistological analysis of the liver samples was also performed in order to evaluate the expression of CD14 molecules in order to connect the inflammatory response and the production of cytokines. Additionally, the changes in the blood cell number and relevant hematological indices were also monitored.

## 2. Results

### 2.1. Blood Analysis

Animals exposed to LPS had a significant increase in the number of neutrophils compared to the control group (*p* < 0.05) (Table 1). On the other hand, in these animals, a decrease in the number of lymphocytes compared to the control (*p* < 0.001) was detected. Treatment with MLT significantly prevented a decrease in the number of lymphocytes when compared to the detected values from animals that received LPS only (*p* < 0.05). In contrast, the number of neutrophils remained unaffected (Table 1). It was also observed that in the blood of rats belonging to the LPS+MEL group, the number of monocytes was slightly elevated when compared to the LPS group (*p* < 0.001). In addition, a decreased number of platelets in rats with endotoxemia was observed compared to the control group (*p* < 0.05) (Table 1). Animals receiving LPS had the values of ALT, AST, and LDH significantly increased compared to the control group, while the treatment with MLT prevented such a significant increase (Table 1).

### 2.2. Hematological Indices

In the blood of rats belonging to the LPS group, a significant increase, compared to the control group (*p* < 0.001), in NLR was found (Table 2). After the combined application of MLT and LPS, a decrease in NLR was detected when compared to the group of animals administered LPS (*p* < 0.05) (Table 2). In the LPS+MEL group, a statistically significant decrease was also observed in PLR in comparison with the values determined for rats from the LPS group (*p* < 0.001) (Table 2).

### 2.3. Liver NF-kB, TNF-α, and IL-6 Levels

In the liver tissue of rats with endotoxemia, the level of NF-κB (254 ± 10.5 pg/mg of proteins) was significantly increased compared to the values of animals belonging to the control group (29.8 ± 9.3 pg/mg of proteins) (*p* < 0.001) (Figure 1A). Oral administration of MLT to rats with endotoxemia led to a significant decrease in NF-κB levels (160.2 ± 57.6 pg/mg of proteins) when compared to the group of animals treated with LPS (254 ± 10.5 pg/mg of proteins) (*p* < 0.001) (Figure 1A).

After LPS application, in the liver tissue, inflammation-related parameters TNF-α and IL-6 were found to be significantly increased in comparison with the values determined for the liver tissue of the control rats (*p* < 0.001) (Figure 1B,C). Application of MLT in combination with LPS prevented an increase in both pro-inflammatory cytokine concentrations when compared to the detected values from animals belonging to the LPS group (Figure 1B,C).

### 2.4. Liver Tissue CD14 Expression

In the control group and group that received MLT, rare CD14-positive cells were detected with semiquantitative scores of 0 and 0.5, respectively (Figure 2A,B). The group of animals that received only LPS had a large number of CD14 cells in the liver tissue with a semiquantitative score of 2.7, mainly present in the spaces around the hepatic artery and vein (Figure 2C). Animals belonging to a group treated with MLT and LPS had a certain number of CD14 cells (Figure 2D) with a semiquantitative score of 1.6, which was higher than in the control group; however, the number was lower than in the group that received only LPS.

## 3. Discussion

During sepsis, acute organ dysfunction as well as generalized inflammatory and procoagulant response to infection occurs. Liver dysfunction was, in the present experiment, proven by an increase in the ALT, AST, and LDH serum levels (Table 1). These include cytokine release, the activation of plasma protein cascade systems, microvascular coagulation, and the activation of neutrophils and monocytes [23]. During this process, neutrophils are activated and increased in number, trying to control the infection. Nonetheless, they can also be harmful to tissue. Due to their dual role, therapeutic goals in sepsis may include the inhibition or activation of neutrophil functions [24]. The results of our study showed that LPS-induced endotoxemia caused a marked increase in the number of neutrophils (Table 1). Neutrophils, as the main circulating phagocytes, are the first and most abundant leukocytes that recruit at the focus of infection during sepsis. It has been proven that sepsis impairs the functioning of the immune system by causing a decrease in the innate immune response [25].

Our research demonstrated that endotoxemia in experimental animals leads to a significant decrease in the number of circulating lymphocytes (Table 1). Many experimental studies in different models of sepsis as well as in patients who died of sepsis with multiple organ damage have shown that sepsis induces a significant loss of lymphocytes through apoptosis. The apoptosis of lymphocytes, which leads to a decrease in their number, is a potential factor involved in immunosuppression and mortality during sepsis. Although the death of the immune system cells can be beneficial because it limits the inflammatory reaction associated with sepsis, the intense apoptosis of lymphocytes leads to a reduced ability of the organism to fight against the bacterial pathogen. The apoptosis of lymphocytes appears to lead to a weakening of the immune response and a predisposition to septic shock and death [26].

This study showed that applying MLT to rats with LPS-induced endotoxemia led to an increase in the number of lymphocytes (Table 1) compared to the LPS group. Melatonin has been shown to enhance the immune response and counteract immunodeficiency states resulting from acute stress, viral and bacterial infections, aging, or treatment with toxic drugs [27,28]. In an in vivo study, MLT supplementation has been shown to reverse immunosuppression induced by stress and glucocorticoids [29] and promote lymphocyte proliferation and antibody production [30].

In the present research, LPS-induced endotoxemia led to a marked decrease in the number of platelets (Table 1). Most clinicians confirm that platelets are involved in the pathogenesis of sepsis since pronounced thrombocytopenia is a common feature of sepsis, and the severity of sepsis is correlated with a decrease in the number of platelets [31]. However, the mechanism of thrombocytopenia in sepsis is not completely clear. Platelet consumption may also play an essential role in patients with sepsis due to the continued generation of thrombin (the most potent platelet activator in vivo). In inflammation-induced coagulation, platelets can be activated directly by endotoxin [32] or pro-inflammatory mediators such as platelet-activating factor [33].

Experimental endotoxemia did not change PLR but caused a significant increase in NLR compared to the control group (Table 2). Recently, PLR and NLR have been monitored in sepsis conditions, and both represent the latest markers of the body’s response to inflammation and predict the prognosis of the disease [34]. Due to an increase in the number of neutrophils and a decrease in the number of lymphocytes, an increase in NLR indicates the poor health of patients with sepsis, who are most often placed in intensive care units. Furthermore, NLR is a reliable marker of early septic shock, serves to monitor the success of therapy, and correlates with the rise of procalcitonin (PCT) [35]. In general, the number of neutrophils and NLR in the blood increases with the progression of sepsis.

The NLR ratio has an induction time of 6 h, and given that it increases faster after the onset of acute physiological stress than the number of leukocytes, it can be concluded that it is a more reliable marker of early septic shock compared to other biomarkers of sepsis. The NLR index has relatively high sensitivity but low specificity and correlates best with PCT elevation [36]. In addition to diagnosing severe sepsis and septic shock, the NLR ratio can also be used to monitor the success of therapy so that in patients who respond well to therapy and recover, the NLR begins to fall within a few days [37]. Our study revealed that treatment with MLT induced a significant diminution of NLR values, detected 12 h after LPS application (Table 2). It has been proven that a high value of the PLR index is significantly associated with increased mortality in patients with sepsis [38]. However, some studies showed that PLR might not be an adequate indicator of sepsis outcome [39]. The results from the present study showed that endotoxemia did not lead to significant changes in PLR (Table 2).

Among the most prominent and best-elucidated processes during endotoxemia is the activation of the NF-κB signaling pathway, which plays a central role in regulating the inflammatory and immune response. In the present research, it was observed that in the liver tissue of rats with endotoxemia, there was a significant increase in the level of NF-κB (Figure 1A). Under physiological conditions, NF-κB sequesters in the cytosol as an inactive homo- or hetero-dimeric form, but when activated, translocates to the nucleus. Genes activated by NF-κB include those encoding the synthesis of various pro-inflammatory cytokines (IL-1β, IL-6, TNF-α), chemokines, adhesion molecules, acute phase proteins (APPs), antimicrobial peptides, inducible nitric oxide synthase, and cyclooxygenase 2. Activation of TLR-4 and CD14 by LPS ultimately results in the nuclear translocation of NF-κB [5,40]. An increase in NF-κB concentration (Figure 1A) in animals exposed to LPS correlated with an increased number of CD14-positive inflammatory cells found in liver tissue (Figure 2C).

Our research demonstrated that the administration of MLT to animals with endotoxemia led to a decrease in NF-κB levels, compared to the LPS group in which its concentration was significantly increased (Figure 1A). Melatonin participates in many critical physiological processes including inflammation reduction [41] and immunoregulation [42]. An earlier study described that MLT blocks the synthesis of NF-κB in HELA cells by stimulating the pro-inflammatory cytokine TNF-α [43]. MLT has also been shown to reduce NF-κB activation in macrophages [44], T cells, liver, and kidney cells [45]. Namely, MLT prevents the translocation of NF-κB to the nucleus and its binding to DNA molecules and consequently reduces the liver inflammatory response after LPS application [46]. It is also important to mention that MLT exerts antioxidant activity [47,48,49] through its antioxidant action; this neurotransmitter protects rats against liver injury induced by endotoxin shock [50] and ischemia/reperfusion damage [51].

The results showed that LPS endotoxemia led to a significant increase in the liver tissue levels of TNF-α and IL-6 (Figure 1B,C). These pro-inflammatory cytokines, TNF-α, IL-1β, and IL-6, are released from activated KCs and are potent stimuli that induce the expression of NGAL (Lipocalin-2) from damaged hepatocytes. Released NGAL induces KCs to release chemokines, which attract neutrophils and monocytes from the blood to the site of infection and further increase inflammation and damage to liver tissue. Attracted neutrophils from their granules release ROS, proteases, and MPO that further worsen the damage and cause hepatocyte necrosis [52]. In high concentrations, TNF-α promotes thrombus formation on the endothelium and lowers blood pressure due to decreased myocardial contractility, dilatation of blood vessels, and increased permeability. IL-6 has been shown to trigger liver regeneration during liver ischemia and to increase the expression of proliferating cell nuclear antigen (PCNA) in the steatotic liver after extensive tissue loss [53]. In addition, IL-6 and activator of transcription protein 3 are involved in organ recovery after liver transplantation [54]; thus, the elevation in the cytokine levels of TNF-α and IL-6 in the liver tissue during endotoxemia can be explained by the tendency of hepatocytes for growth and proliferation during their damage.

Previous studies have shown that MLT inhibits the LPS-stimulated production of TNF-α, IL-1β, IL-6, IL-8, and IL-10 in cells through a mechanism involving a reduction in the activation level of the NF-κB signaling pathway [55]. MLT also reduces the synthesis of IL-6 and NO during LPS-induced endotoxemia by blocking the NF-κB signaling pathway, thereby blocking its nuclear translocation and DNA binding of the NF-κB p50 subunit and suppressing STAT-1 signaling [56]. Additionally, MLT reduces TLR-3-mediated TNF-α and iNOS expression by inhibiting NF-κB activation in respiratory virus-infected RAW 264.7 macrophages [57]. Melatonin has been shown to reduce methamphetamine-induced increases in the TNF-α, IL-1β, and IL-6 levels in rat HAPI glial cells [58] and beta-amyloid-induced TNF-α and IL-6 overproduction [59]. The results of our research showed that the administration of MLT to animals with endotoxemia led to a decrease in the level of cytokines TNF-α and IL-6 in the liver tissue compared to the group of animals administered LPS, in which their concentration was significantly increased (Figure 1B,C). These findings are in accordance with the results of the previously mentioned research. Previous research has also shown that MLT blocks the excessive production of pro-inflammatory cytokines, especially TNF-α, and increases the level of IL-10 [21].

## 4. Materials and Methods

### 4.1. Chemicals and Reagents

Lipopolysaccharide (from Escherichia coli O111:B4; LPS) and melatonin (MLT) were purchased from Sigma (St. Louis, MO, USA). Ketamine (Ketamidor 10%), used as a general anesthetic, was obtained from Richter Pharma AG (Wels, Austria). All other used chemicals were of analytical grade purity.

### 4.2. Animals and Housing

Experiments were performed on healthy male Wistar Albino rats (7 weeks old, weighing 150 to 200 g) obtained from the Vivarium of the Institute of Biomedical Research, Faculty of Medicine, University of Niš, Serbia. The animals (the total number of animals used in the study was 24) were kept under conventional laboratory conditions, with a temperature of 22 ± 2 °C, relative humidity of 50 ± 5%, and a 12/12 h light/dark cycle. Animals were given ad libitum access to water and standard, commercial laboratory food. The local Ethics Committee approved the current study. All experimental procedures were performed in accordance with the ethical regulations of the Helsinki and European Community guidelines for the ethical handling of laboratory animals (EU Directive of 2010; 2010/63/EU) as well as those given by the laws of the Republic of Serbia (Decision number: 323–07–01762/2021–01).

### 4.3. Experimental Design

Animals were randomly divided into four groups, each consisting of six rats, except for Group III, which consisted of 12 animals (these calculations were made based on the pilot experiment). In order to induce sepsis, LPS was applied as a single intraperitoneal dose of 10 mg/kg body weight (b.w.) [7]. The solution of MLT was prepared prior to LPS application at a dose of 50 mg/kg b.w. [7]. Rats were treated following a pre-determined schedule: Group I (Vehicle)—8% ethanol in saline in a dose of 10 mL/kg b.w. by oral gavage; Group II (MLT)—single dose of MLT (50 mg/kg b.w.) administered by oral gavage; Group III (LPS)—single intraperitoneal injection of LPS at a dose of 10 mg/kg b.w.; Group IV (LPS+MLT)—single dose of MLT (50 mg/kg b.w.; p.o.), followed by a single dose of LPS (10 mg/kg b.w.; i.p.). During the experiment, 50% of the animals died only in the group of rats treated with LPS (Group III), while no lethal outcomes were noted in the other groups.

Twelve hours after the treatments, the animals were sacrificed with an overdose of ketamine, after which the blood samples and liver tissue were taken for serum analysis, hematological determinations, and immunohistochemical analyses.

### 4.4. Serum and Blood Hematological Analysis

Blood samples were obtained with and without an anticoagulant from the animals by a cardiac puncture, and the sample was further used to evaluate the serum ALT, AST, and LDH activities (Olympus AU680) and hematological parameters (DxH 500-Beckman Coulter). Additionally, from the obtained results, the neutrophils to lymphocytes ratio (NLR) and platelets to lymphocytes ratio (PLR) were calculated.

### 4.5. Liver Tissue Homogenate Preparation

Briefly, liver tissue specimens were finely cut into small pieces and homogenized (10%, *w*/*v*) in ice-cold distilled water. Tissue parameters were determined in clear supernatants obtained after centrifugation (3210 g, 15 min, 4 °C). Protein content in the supernatants was determined by Lowry’s method [60], using the standard curve constructed with bovine serum albumin as a standard.

### 4.6. ELISA Assays

Commercial kits for determinations of the NF-κB level (NF-kappa-B-activating protein ELISA Kit, Wuhan Fine Biotech, Wuhan, China; ER0510) and the tissue concentrations of TNF-α (Rat TNF alpha ELISA Kit Abcam, Boston, MA, USA; ab236712) and IL-6 (Quantikine ELISA Rat IL-6, R&DSystems, Minneapolis, MN, USA; R6000B) were used and assays were conducted according to the manufacturer’s instructions. The concentrations of the determined parameters were presented as pg/mg of proteins.

### 4.7. Liver Tissue Immunohistochemical Analyses

Rat liver tissue samples were fixed in formaldehyde solution (10%, *w*/*v*), dehydrated with ethanol solutions of differing concentrations (50–100%, *v*/*v*), embedded in paraffin molds, cut into 4-μm thick sections, and stained for CD14 expression. Immunohistochemical staining was conducted following previously described standard procedures [61]. The standard procedure of antigen retrieval (citrate buffer) and endogenous peroxidase blockage (using 3% hydrogen peroxide) was performed prior to an overnight incubation with the primary rabbit polyclonal anti-CD14 antibody (1:60) (Thermo Fisher Scientific, Waltham, MA, USA) in a moist chamber. The visualization was effectuated using diaminobenzidine and counterstained with Mayer’s hematoxylin. Pathohistological analysis of the stained slides was performed on a light microscope (Olympus BX43, Olympus Corporation, Tokyo, Japan) [61]. Immunohistochemical staining was semi-quantitatively graded as: absent/no staining (0), trace (0.5), mild (1), moderate (2), and strong positive staining (3) [61].

### 4.8. Statistical Analysis

Results were expressed as the mean values ± standard deviation (SD). Statistically significant differences were determined by one-way analysis of variance (ANOVA) followed by Tukey’s post hoc test for multiple comparisons (GraphPad Prism version 5.03, San Diego, CA, USA). Probability values (*p*) ≤ 0.05 were considered to be statistically significant.

## 5. Conclusions

The results of the present study proved with further evidence that melatonin could act as a potential modulator of sepsis induced by lipopolysaccharide. The impact of melatonin is visible on circulating blood cells, whose number and indices were maintained near physiological values. Apart from that, melatonin prevented an increase in NF-kB, TNF-α, and IL-6, suggesting its importance in regulating inflammation at the molecular level. Finally, melatonin prevented an increase in CD14-positive inflammatory cell infiltration, directly associated with its function as a cytokine production modulator.

## Figures and Tables

**Figure 1 ijms-25-00455-f001:**
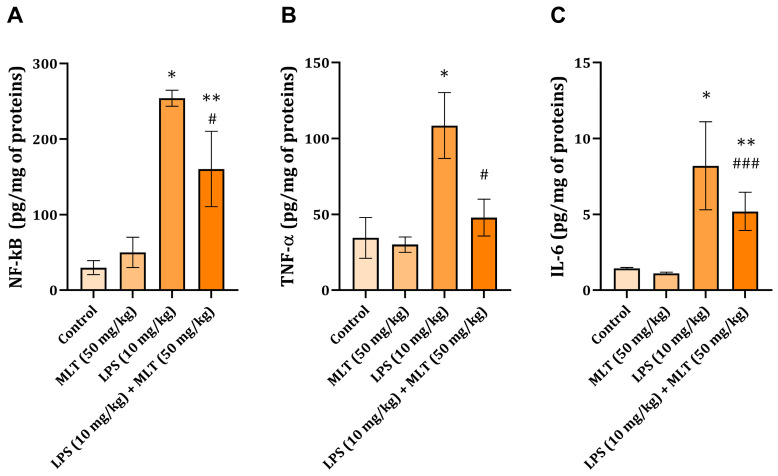
The effects of melatonin (MLT) on LPS-induced changes in NF-κB level (**A**), TNF-α (**B**), and IL-6 (**C**) concentrations (given in pg/mg of proteins) in the liver tissue of experimental animals. The data are presented as mean ± SD (*n* = 6). The comparison was conducted using one-way ANOVA followed by Tukey’s post hoc test. ** *p* < 0.01, * *p* < 0.001 vs. control; # *p* < 0.001, ### *p* < 0.05 vs. LPS treated animals.

**Figure 2 ijms-25-00455-f002:**
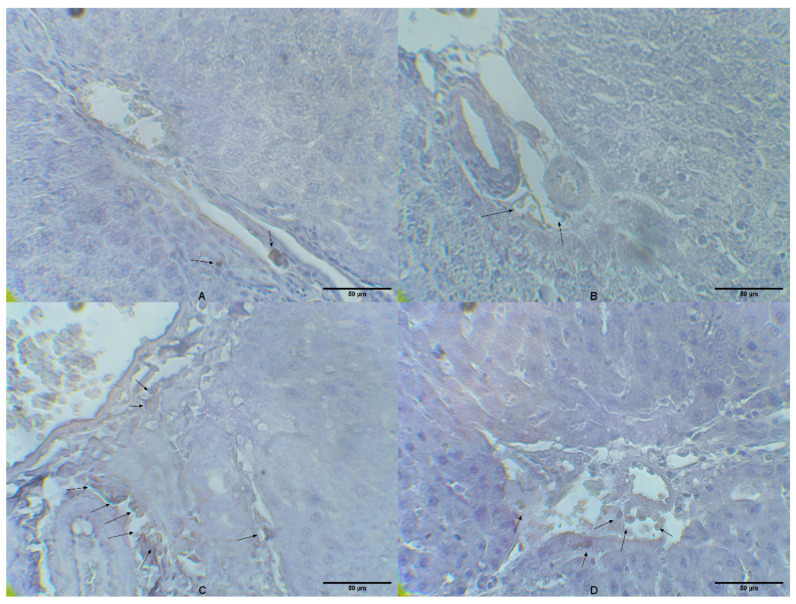
Expression of CD14 in the liver tissue of rats belonging to different experimental groups (magnification ×400). (**A**) Liver of control group rats with rare CD14 expression; (**B**) liver of MLT treated rats with rare CD14 expression; (**C**) liver of LPS treated rats with frequent perisinusoidal CD14 expression; (**D**) liver of LPS+MLT treated rats with occasional perisinusoidal CD14 expression. Cells expressing CD14 are marked with arrows.

**Table 1 ijms-25-00455-t001:** Hematological and biochemical parameters in the blood of animals belonging to different experimental groups.

Parameter	Control	Melatonin	LPS	LPS+MLT
Red blood cell number (×10^12^/L)	7.53 *±* 0.44	7.20 *±* 0.2	7.31 *±* 0.61	6.97 *±* 0.73
Hematocrit	0.42 *±* 0.02	0.41 *±* 0.01	0.40 *±* 0.03	0.39 *±* 0.04
Hemoglobin (g/L)	141.9 *±* 6.2	136.8 *±* 1.5	142.1 *±* 7.4	137.9 *±* 5.1
MCV (fl)	55.7 *±* 1.52	57.6 *±* 2.35	55.5 *±* 1.98	57.3 *±* 1.95
MCH (pg)	18.85 *±* 0.41	19.03 *±* 0.59	19.23 *±* 0.72	19.13 *±* 0.48
MCHC (g/L)	338 *±* 7.3	330 *±* 7.9	347 *±* 12.8	334 *±* 5.3
RDW (%)	20.9 *±* 0.94	20.03 *±* 1.15	20.31 *±* 0.93	19.66 *±* 0.75
White blood cell number (×10^9^/L)	6.55 *±* 3.22	7.94 *±* 1.4	3.95 *±* 1.73	5.68 *±* 1.59
Neutrophiles number (×10^9^/L)	0.77 *±* 0.53	0.67 *±* 0.24	1.4 *±* 0.84 ***	1.31 *±* 0.45
Lymphocytes number (×10^9^/L)	4.84 *±* 1.83	6.39 *±* 0.85 *	2.37 *±* 0.58 *	3.61 *±* 0.83 ^#^
Monocytes number (×10^9^/L)	0.11 *±* 0.07	0.17 *±* 0.06	0.19 *±* 0.06	0.25 *±* 0.02 ^#^
Eosinophiles number (×10^9^/L)	0.025 *±* 0.018	0.026 *±* 0.005	0.041 *±* 0.014	0.07 *±* 0.064
Basophile number (×10^9^/L)	0.041 *±* 0.026	0.043 *±* 0.005	0.073 *±* 0.052	0.075 *±* 0.046
Platelets number (×10^9^/L)	313 *±* 89	437 *±* 159	168 *±* 74 ***	126 *±* 21
MPV (fl)	7.98 *±* 1.00	8.13 *±* 0.61	9.09 *±* 1.49	10.43 *±* 1.37
ALT (U/L)	50.1 *±* 7.2	54.3 *±* 0.9	281.4 *±* 76.2 *	133 *±* 27.3 ^#^
AST (U/L)	162 *±* 18.9	158 *±* 24.3	663 *±* 119.7 *	571 *±* 211 *
LDH (U/L)	2008 *±* 114	2004 *±* 298	2477 *±* 446 ***	2171 *±* 445 ^##^

Data are shown as mean ± SD (*n* = 6). The comparison was conducted using one-way ANOVA, followed by Tukey’s post hoc test. * *p* < 0.001, *** *p* < 0.05 vs. control; ^#^
*p* < 0.001, ^##^ *p* < 0.01 vs. LPS treated animals.

**Table 2 ijms-25-00455-t002:** Hematological indices (platelet to lymphocyte and neutrophil to lymphocyte ratio) for animals belonging to different experimental groups.

Parameter	Control	Melatonin	LPS	LPS+MLT
Platelet to lymphocyte ratio (PLR)	72.2 ± 49.8	73.2 ± 49.9	77.4 ± 45	39.4 ± 11.7 ^#^
Neutrophil to lymphocyte ratio (NLR)	0.15 ± 0.07	0.11 ± 0.04	0.59 ± 0.32 *	0.35 ± 0.07 ^###^

Data are shown as mean ± SD (*n* = 6). The comparison was conducted using one-way ANOVA, followed by Tukey’s post hoc test. * *p* < 0.001 vs. control; ^#^ *p* < 0.001, ^###^ *p* < 0.05 vs. LPS treated animals.

## Data Availability

The data are available upon reasonable request from the corresponding author.

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
