# Peer review of "Melatonin as a Promising Anti-Inflammatory Agent in an In Vivo Animal Model of Sepsis-Induced Rat Liver Damage"

_ijms, 2023, doi:10.3390/ijms25010455_

Round 1

Reviewer 1 Report

Comments and Suggestions for Authors

The present experimental work describes the anti-inflammatory role of melatonin in a rat animal model of hepatic sepsis induced by LPS. The authors conclude that melatonin decreases the anti-inflammatory effects of LPS. The anti-inflammatory effects of melatonin are widely known, not only in the liver but also in similar organs such as pancreatic tissue (for example: Carrasco et al. Cell Biochem Funct 31, 2013), so this work does not represent an advance in the knowledge that exists in this regard.

In any case, if the editor deems it appropriate, the work may be considered for publication with some minor modifications.

1-Include the body weight of the animal in the doses of LPS and melatonin. That is, the doses of LPS and MLT were 10 and 50 mg/Kg b.w. (body weight), respectively.

2- The experimental protocol indicates that the MLT was administered orally prior to the intraperitoneal injection of LPS. That is, the administration was practically simultaneous, without pretreatment with melatonin. It is difficult to understand that the effects of melatonin manifest so quickly. Normally, retreatment with melatonin will be required for a few hours or days to be effective. My question to the authors is if they have previous results with pretreatment with melatonin for 2-4-6-12 or 24 hours. In relation to this, they should remove the term "pretreatment with melatonin" in the entire text, including the abstract.

3- Finally, figures 1, 2 and 3 of the liver levels of NFKb, TNF-alpha and IL-6 can be grouped into a single figure.

Author Response

Dear Editor,

I am resubmitting the manuscript entitled: “Melatonin as a promising anti-inflammatory agent in in vivo animal model of sepsis-induced rat liver damage“ for consideration of publication after comments suggested by reviewers have been addressed.

The manuscript has been amended according to all of the reviewers’ comments and suggestions. All the changes made to the text are highlighted in yellow throughout the manuscript document. Point-by-point answers to specific reviewers’ comments are given below.

If any further matter needs to be resolved, please, do not hesitate to contact me.

Sincerely,

Professor Daniela Benedeto Stojanov

Reviewer(s)' Comments to Author:

Reviewer 1:

The present experimental work describes the anti-inflammatory role of melatonin in a rat animal model of hepatic sepsis induced by LPS. The authors conclude that melatonin decreases the anti-inflammatory effects of LPS. The anti-inflammatory effects of melatonin are widely known, not only in the liver but also in similar organs such as pancreatic tissue (for example: Carrasco et al. Cell Biochem Funct 31, 2013), so this work does not represent an advance in the knowledge that exists in this regard.

In any case, if the editor deems it appropriate, the work may be considered for publication with some minor modifications.

1-Include the body weight of the animal in the doses of LPS and melatonin. That is, the doses of LPS and MLT were 10 and 50 mg/Kg b.w. (body weight), respectively.

Answer: Thank you for your suggestion. We have included the abbreviation b.w.t in the corrected manuscript text, where it was necessary.

2- The experimental protocol indicates that the MLT was administered orally prior to the intraperitoneal injection of LPS. That is, the administration was practically simultaneous, without pretreatment with melatonin. It is difficult to understand that the effects of melatonin manifest so quickly. Normally, retreatment with melatonin will be required for a few hours or days to be effective. My question to the authors is if they have previous results with pretreatment with melatonin for 2-4-6-12 or 24 hours. In relation to this, they should remove the term "pretreatment with melatonin" in the entire text, including the abstract.

Answer: Indeed, MLT was applied simultaneously with LPS, thus it could not be considered as pre-treatment. We have corrected this through the manuscript text. We have not investigated other treatment regimens with MLT in the current experimental setting, nor in our pilot experiments. The idea of how MLT could act and is the treatment regimen changing the outcome of the experiment is a nice idea that we might incorporate in our future studies on this topic. These are initial results from our studies, and in future we plan to examine the effect of MLT on different organs to see its general action when administered with LPS at the same time.

3- Finally, figures 1, 2 and 3 of the liver levels of NFKb, TNF-alpha and IL-6 can be grouped into a single figure.

Answer: We have grouped figures 1, 2 and 3 of the liver levels of NF-kB, TNF-alpha and IL-6 concentrations into a single figure.

Reviewer 2 Report

Comments and Suggestions for Authors

The manuscript entitled “Melatonin as a promising anti-inflammatory agent in in vivo 2 animal model of sepsis-induced rat liver damage” describes an interesting study about how melatonin may have anti-inflammatory properties in a rat sepsis model.

It is a well-written and executed manuscript the discussion is appropriate and the results are in accordance with the experiments carried out.

The introduction clarifies the background and objectives of the research. The conclusion is reasonable and provides appropriate knowledge for future research.

Nevertheless, I have a major concern:

The authors need to justify why they only choose 6 animals per group instead of a larger number. And how this may have affected to significance, moreover if 3 animals died in the LPS group, this will definitively affect the validation of statistics. Therefore, figure 3 may also be affected by this fact.

 I consider that this concern is addressed, this manuscript will have a good scientific quality, with results that are of interest great interest. 

Author Response

Dear Editor,

I am resubmitting the manuscript entitled: “Melatonin as a promising anti-inflammatory agent in in vivo animal model of sepsis-induced rat liver damage“ for consideration of publication after comments suggested by reviewers have been addressed.

The manuscript has been amended according to all of the reviewers’ comments and suggestions. All the changes made to the text are highlighted in yellow throughout the manuscript document. Point-by-point answers to specific reviewers’ comments are given below.

If any further matter needs to be resolved, please, do not hesitate to contact me.

Sincerely,

Professor Daniela Benedeto Stojanov

Reviewer(s)' Comments to Author:

Reviewer 2:

The manuscript entitled “Melatonin as a promising anti-inflammatory agent in in vivo 2 animal model of sepsis-induced rat liver damage” describes an interesting study about how melatonin may have anti-inflammatory properties in a rat sepsis model.

It is a well-written and executed manuscript the discussion is appropriate and the results are in accordance with the experiments carried out.

The introduction clarifies the background and objectives of the research. The conclusion is reasonable and provides appropriate knowledge for future research.

Nevertheless, I have a major concern:

The authors need to justify why they only choose 6 animals per group instead of a larger number. And how this may have affected to significance, moreover if 3 animals died in the LPS group, this will definitively affect the validation of statistics. Therefore, figure 3 may also be affected by this fact.

I consider that this concern is addressed, this manuscript will have a good scientific quality, with results that are of interest great interest.

Answer: Thank you very much for your mention that our present manuscript has good scientific quality and results that are of interest. In our statistics analysis, we used one-way ANOVA followed by Tukey’s post hoc test for multiple comparisons. Indeed we reported that 50% of animals died in the group receiving LPS, which is in accordance with some previous studies. Since we have knowledge of this phenomenon and some experience from pilot experiments, we expected such thing in the group receiving only LPS. Thus this group consisted out of 12 animals, and half of them died after the LPS injection. Remaining 6 animals were included in the analysis. We mended this in the materials and methods section.

Reviewer 3 Report

Comments and Suggestions for Authors

The authors used lipopolysaccharide (LPS) to induce sepsis in rats, a condition that causes systemic inflammation and organ damage. They measured the levels of pro-inflammatory cytokines, NF-κB, and CD14 in the liver tissue and blood of the rats. The authors tested the effects of melatonin (MLT), a hormone with anti-inflammatory and antioxidant properties, on the sepsis model. They find that MLT reduces the levels of pro-inflammatory cytokines, NF-κB, and CD14 in the liver tissue and blood of the rats, and prevents the decrease of lymphocytes and platelets. The authors concluded that MLT acts as a potential modulator of sepsis by attenuating the inflammatory response and the liver damage induced by LPS. They suggest that MLT could be a beneficial adjunctive drug for sepsis treatment.

1.          What is the mechanism of action of melatonin in modulating the immune response and inflammation in sepsis? How does it interact with the TLR-4 and CD14 receptors?

2.   How did you determine the optimal dose and timing of melatonin administration for the rats? Did you test different doses or time intervals?

3.   How did you perform the immunohistochemical analysis of CD14 expression in the liver tissue? What antibody and staining protocol did you use? How did you quantify the CD14-positive cells?

4.      How did you assess the liver function and damage in the rats? Did you measure any biochemical markers or histological parameters of liver injury?

5.          How did you control for the possible effects of ethanol, which was used as a vehicle for melatonin, on the liver and the immune system of the rats?

6.          Figure 4 is not labeled properly or is too small to read. I recommend revising these figures to make them more clear and readable.

7.          Please provide the scale bars of Figure 4.

8.         How do you compare your results with other studies that have investigated the effects of melatonin on sepsis or liver damage? What are the similarities and differences?

Author Response

Dear Editor,

I am resubmitting the manuscript entitled: “Melatonin as a promising anti-inflammatory agent in in vivo animal model of sepsis-induced rat liver damage“ for consideration of publication after comments suggested by reviewers have been addressed.

The manuscript has been amended according to all of the reviewers’ comments and suggestions. All the changes made to the text are highlighted in yellow throughout the manuscript document. Point-by-point answers to specific reviewers’ comments are given below.

If any further matter needs to be resolved, please, do not hesitate to contact me.

Sincerely,

Professor Daniela Benedeto Stojanov

Reviewer(s)' Comments to Author:

Reviewer 3:

The authors used lipopolysaccharide (LPS) to induce sepsis in rats, a condition that causes systemic inflammation and organ damage. They measured the levels of pro-inflammatory cytokines, NF-κB, and CD14 in the liver tissue and blood of the rats. The authors tested the effects of melatonin (MLT), a hormone with anti-inflammatory and antioxidant properties, on the sepsis model. They find that MLT reduces the levels of pro-inflammatory cytokines, NF-κB, and CD14 in the liver tissue and blood of the rats, and prevents the decrease of lymphocytes and platelets. The authors concluded that MLT acts as a potential modulator of sepsis by attenuating the inflammatory response and the liver damage induced by LPS. They suggest that MLT could be a beneficial adjunctive drug for sepsis treatment.

  1. What is the mechanism of action of melatonin in modulating the immune response and inflammation in sepsis? How does it interact with the TLR-4 and CD14 receptors?

Answer: Thank you for your constructive suggestion. In the section Introduction we tried to clarify the modulation of TLR/NF-kB signaling pathway modulation by melatonin in the sepsis conditions, by citing the following references (15, 16, 18). Our suggestion based on the literature reviewe and based on the results of our study that MLT interreacts with the cascade of events that follow TLR-4 and CD14 receptor activation. We did not find any reference regarding this direct interaction, and if the reviewer has some suggestions we are more than glad to include them in the manuscript.

  1. How did you determine the optimal dose and timing of melatonin administration for the rats? Did you test different doses or time intervals?

Answer: Thank you for your comment. The dose of MLT used in the experiment is based on extensive literature review and more than a decade experience with MLT in in vivo experiments. This is just first experiment with LPS and MLT and future ones, based on these studies, will focus on different treatment regimens and doses.

  1. How did you perform the immunohistochemical analysis of CD14 expression in the liver tissue? What antibody and staining protocol did you use? How did you quantify the CD14-positive cells?

Answer: Antibody type, with correction of the producer, was added to the protocol and some details about the procedure which are necessary for the repetition. Also, semi-quantitative procedure for the CD14 cell positivity was added and the results text was extended with the obtained results.

  1. How did you assess the liver function and damage in the rats? Did you measure any biochemical markers or histological parameters of liver injury?

Answer: Thank you for your very constructive question. We did perform the biochemical analysis but we did not want to burden the text, since the action of both MLT and LPS + MLT has been proven. We added the values for ALT, AST and LDH in the Table 1 and have mentioned them through the text.

  1. How did you control for the possible effects of ethanol, which was used as a vehicle for melatonin, on the liver and the immune system of the rats?

Answer: The mode for MLT dilution and the usage of this vehicle has been previously proven to be without any effect, especially since the volume of MLT administered (0.1 ml) and the percent of ethanol is very low. This was done in many previous experiments, thus we did not include such control in the current one.

  1. Figure 4 is not labeled properly or is too small to read. I recommend revising these figures to make them more clear and readable.

Answer: The images are of magnification where structures of liver with some cells which are positive. Larger magnification would not give a clear insight how the diffuse positivity is present in the tissue. Addition of the semi-quantitative score improved the understanding of the histopathological findings.

  1. Please provide the scale bars of Figure 4.

Answer: We have provided the scale bars for Figure 4.

  1. How do you compare your results with other studies that have investigated the effects of melatonin on sepsis or liver damage? What are the similarities and differences?

Answer: Thank you for your suggestion. In the section Discussion (old version of the Manuscript) we try to explain our obtained results in more details, compared with the other research groups findings. Our study confirmed that melatonin application in sepsis induced many similarities which are highly correlated with results from scientific article in this field of investigation, especially, when we talk about the hematological parameters ratio or pro-inflammatory levels in liver tissue.

Round 2

Reviewer 2 Report

Comments and Suggestions for Authors

My major concern was correctly addressed. 

I recommend to accept it in present form.